# Limited Utility of SIRS Criteria for Identifying Serious Infections in Febrile Young Infants

**DOI:** 10.3390/children8111003

**Published:** 2021-11-03

**Authors:** Osamu Nomura, Yoshihiko Morikawa, Takaaki Mori, Yusuke Hagiwara, Hiroshi Sakakibara, Yuho Horikoshi, Nobuaki Inoue

**Affiliations:** 1Division of Pediatric Emergency Medicine, Tokyo Metropolitan Children’s Medical Center, Tokyo 183-8561, Japan; takaakimori001019@gmail.com (T.M.); yusukehagiwara-tky@umin.ac.jp (Y.H.); nobuinoue@me.com (N.I.); 2Department of Emergency and Disaster Medicine, Hirosaki University, Aomori 036-8562, Japan; 3Clinical Research Support Center, Tokyo Metropolitan Children’s Medical Center, Tokyo 183-8561, Japan; yoshihiko_morikawa@tmhp.jp (Y.M.); hiroshi_sakakibara@tmhp.jp (H.S.); 4Department of General Pediatrics, Tokyo Metropolitan Children’s Medical Center, Tokyo 183-8561, Japan; yuho74@hotmail.com; 5Division of Infectious Diseases, Department of Pediatrics, Tokyo Metropolitan Children’s Medical Center, Tokyo 183-8561, Japan; 6Bureau of International Health Cooperation, Department of Human Resources and Development, National Center for Global Health and Medicine, Tokyo 183-8561, Japan

**Keywords:** febrile young infants, sepsis, systematic inflammatory response syndrome, serious infection

## Abstract

(1) Background: Young infants have a high risk of serious infection. The Systematic Inflammatory Response Syndrome (SIRS) criteria can be useful to identify both serious bacterial and viral infections. The aims of this study were to evaluate the diagnostic performance of the SIRS criteria for identifying serious infections in febrile young infants and to identify potential clinical predictors of such infections. (2) Methods: We conducted this prospective cohort study including febrile young infants (aged < 90 days) seen at the emergency department with a body temperature of 38.0 °C or higher. We calculated the diagnostic performance parameters and conducted the logistic regression analysis to identify the predictors of serious infection. (3) Results: Of 311 enrolled patients, 36.7% (*n* = 114) met the SIRS criteria and 28.6% (*n* = 89) had a serious infection. The sensitivity, specificity, positive predictive value, and positive likelihood ratio of the SIRS criteria for serious infection was 45.9%, 69.4%, 43.5%, 71.4%, 1.5, and 0.8, respectively. Logistic regression showed that male gender, body temperature ≥ 38.5 °C, heart rate ≥ 178 bpm, and age ≤ 50 days were significant predictors. (4) Conclusions: The performance of the SIRS criteria for predicting serious infections among febrile young infants was poor.

## 1. Introduction

Young infants (aged < 90 days) have a high risk of developing serious infections (SIs) due to their greater susceptibility to pathogens and problems with clinical examination that may cause potential infections to be overlooked [1,2]. Traditionally, the clinical care of febrile young infants has focused on identifying serious bacterial infections requiring early intervention with antibiotics. Diagnosing viral infections was emphasized less due to the self-limiting clinical course of most viral infections. However, recent studies have shown the importance of diagnosing viral infections in the emergency department (ED) due to the fact that several types of viral infection can cause respiratory failure, septic shock, or central nervous system dysfunction requiring intensive care, possibly resulting in unfavorable outcomes in young infants [3,4,5,6]. This has led some researchers to argue for the need to diagnose viral infections in infants [7,8,9]. Although several, point-of-care, rapid viral tests are available to detect respiratory syncytial virus (RSV), human metapneumovirus, and influenza virus infections in the outpatient setting, emergent viral infections that may cause septic shock, such as a parechovirus or enterovirus infection, cannot be diagnosed by the point-of-care testing. In addition, it is arguable to screen parechovirus and or enterovirus in all febrile young infants as not all of the patients with these viruses become septic. Several, low-risk criteria and clinical prediction models have been developed to screen for serious bacterial infections in febrile young infants [10,11]. Although these diagnostic tools have expedited clinicians’ decision making and helped to improve the use of resources in pediatric acute care [12,13], clinicians may still misdiagnose serious viral infections if they rely only on methods used to diagnose bacterial infections [14,15].

The concept of sepsis is useful in this context because it is defined by the systemic inflammatory response syndrome (SIRS) in the presence of a suspected or proven infection, i.e., patients with sepsis can be suffering either from a serious bacterial or viral infection [16,17]. The SIRS criteria in accordance with the International Pediatric Sepsis Conference Guidelines have been suggested as a screening strategy for identifying pediatric patients at risk for septic shock. The pediatric SIRS definitions require the presence of at least two of the following, one of which should be abnormal temperature or white blood cell (WBC) count: core body temperature (<36 or >38.5), tachycardia or bradycardia, tachypnea, elevated WBC for age [18,19]. As discussed earlier, infants younger than 90 days old are susceptible to SI and using the ‘sepsis-approach’ to predict serious bacterial and viral infections in these patients is logical. While the sepsis guidelines for adult patients no longer recommend using the SIRS to screen the sepsis patients, no studies have directly investigated the diagnostic utility of the SIRS criteria for identifying severe infections in young infants. We herein examined the diagnostic utility of the SIRS criteria evaluated at ED admission for identifying SI in febrile young infants. Furthermore, we aimed to explore the potential clinical parameters for predicting SI in these patients.

## 2. Materials and Methods

### 2.1. Study Setting

Tokyo Metropolitan Children’s Medical Center is a pediatric tertiary care center in Tokyo, Japan with about 38,000 annual pediatric visits to the ED. While the initial management of patients is usually provided by pediatric or emergency medicine residents, board-certificated emergency medicine physicians or pediatricians supervise patient care [20].

### 2.2. Study Design and Selection of Participants

We conducted this prospective cohort study from 1 August 2014 through 30 September 2016. This study was initiated before the updated clinical policy and guidelines for febrile young infants of the American College of Emergency Physicians and American Academy of Pediatrics were published [1,2]. All young infants (aged < 90 days) who visited our ED with a core body temperature of 38.0 °C or higher at triage were included. Patients who received medical care at any hospitals 72 h before the presentation to our ED were excluded because their prior treatment might have affected the data of these patients.

We also followed up on all patients in our cohort by speaking with their guardians via telephone to assess for 28-day mortality.

### 2.3. Data Collection

We collected information on patient demographics including patient age, sex, history, vital signs, physical examination findings, laboratory examination results, definitive diagnosis, and outcomes. The vital sign and laboratory variables related to SIRS criteria (e.g., WBC counts) were evaluated at admission to the ED. The diagnosis data of the patients were obtained during the clinical course of the disease and finally defined at the time of telephone follow-up for assessing the 28-day outcome.

### 2.4. Definition

For this study, we adopted the definition of SIRS provided by the International Pediatric Sepsis Conference Guideline [18]. In our measurement of the vital signs, we adopted the minimum values for the heart and respiratory rate within the first 30 min of patient care to eliminate the influence of crying or agitation. If the patients received any treatment including intravenous fluids or medications within 30 min of the measurements, we adopted the minimum value of the vital signs before the interventions were given.

We defined a serious infection in the following manner by applying the definition used in previous studies that covers both bacterial and viral infections [4,21,22].

(1)Septicemia (including bacteremia and viremia): pathogenic bacteria isolated from a blood culture or a pathogenic virus detected by real-time PCR.(2)Meningitis: identification of bacteria or a virus in cerebrospinal fluid.(3)Pneumonia: infiltrate on chest X-ray.(4)Soft tissue infection: acute suppurative inflammation of subcutaneous tissue.(5)Urinary tract infection: >positive urine culture with a single species of pathogen and systemic symptoms such as fever.(6)Bronchiolitis due to RSV or human metapneumovirus requiring emergency intervention and/or evidence of organ failure.

We used the above definition of bronchiolitis to select severe cases. ‘Emergency intervention’ was defined as a clinical state requiring emergency supplemental oxygen or fluid resuscitation of more than 20 ml/kg in the ED indicative of respiratory distress/failure or compensatory/hypotensive shock according to the pediatric sepsis guidelines [23]. Other evidence of organ failure included altered mental status, renal failure, and hematologic and/or hepatic failure, according to the pediatric Sequential Organ Failure Assessment (SOFA) score [24,25].

Viruses were detected by a rapid antigen test (commercial immunochromatography kit) or real-time PCR. Patients with lower respiratory infections were tested for RSV, human metapneumovirus, and the influenza virus by rapid antigen test. Viremia and/or central nervous system infections due to parechovirus or enterovirus were diagnosed by routinely performing real-time PCR for all hospitalized patients in whom the focus of infection was unknown.

### 2.5. Analysis

Data were analyzed using the SPSS software package, version 23 (Armonk, NY: IBM Corp). Standard descriptive statistics were reported with the mean and SD for continuous variables and frequency and percentage for categorical variables. The chi-squared and Student’s t-test were conducted for statistical analysis, and a two-sided *p* < 0.05 was considered statistically significant. Diagnostic performance parameters including sensitivity, specificity, positive predictive value (PPV), negative predictive value (NPV), and positive/negative likelihood ratio (LR+/−) of the SIRS criteria were calculated for SI. Multivariate logistic regression analysis was performed to identify the predictors of SI based on an extensive review of past studies and their clinical implications. To broaden their clinical application, continuous variables were converted into categorical variables by determining each variable’s cut-off point prior to logistic regression, and the optimal cutoff point for each variable was calculated using the receiver operating characteristic curve for SI.

## 3. Results

### 3.1. Patient Characteristics

Of the 327 eligible patients, sixteen were excluded due to a previous visit to our ED, receiving some form of intervention or insufficient data (Figure 1). The remaining 311 patients (167 boys (53.7%); mean age (days) (SD), 52.6 (23.7)) were included. One hundred-fourteen (36.7%) patients met the SIRS criteria at ED admission, and 89 (28.6%) patients had a serious infection. Among the patients with SI, 41 had a serious bacterial infection and 48 had a serious viral infection (Table 1).

Of all the patients, 177 (56.9%) were hospitalized in a ward, and seven (2.3%) were admitted to the pediatric intensive care unit. No patient died within 28 days following the first visit to our ED.

### 3.2. Performance of the SIRS Criteria

The sensitivity, specificity, PPV, NPV, LR+, and LR− of the SIRS criteria for SI was 47.2%, 68.0%, 37.2%, 76,3%, 1.5 (95% confidence interval (CI), 1.1–2.0), and 0.8 (95%CI, 0.6–0.9), respectively (Table 2).

### 3.3. Predictive Factors for Serious Infection

Patients with a SI were slightly younger than patients without a SI, but the difference was not statistically significant (49.2 days [25.2] vs. 54.1 days [23.0], *p* = 0.101). The proportion of males was significantly higher among SI patients than among non-SI patients (68.5% vs. 47.7%, *p* = 0.001). Regarding the vital signs, SI patients presented a significantly higher body temperature (38.7 °C [0.5] vs. 38.5 °C [0.4], *p* = 0.004) and tachycardia (175.6 beats per minute: bpm [11.2] vs. 167.9 bpm [18.4], *p* = 0.001). Although the respiratory rate was similar between the two groups, the patients with SI showed lower oxygen saturation (97.9 % [3.2] vs. 98.5 % [1.6], *p* = 0.015) than the patients without SI. The majority of patients in the SI group showed poor perfusion in their extremities (51.7% vs. 36.5%, *p* = 0.016) and prolonged capillary refilling time (31.5% vs. 19.4%, *p* = 0.025). While there was no difference in the WBC count between the groups, the percentage of lymphocytes was significantly smaller (39.3% vs. 44.5%, *p* = 0.008) in the SI group. No differences were found in the percentage of monocytes and the venous lactate value between the groups (Table 3).

Among these predictors, we selected the variables of age, gender, heart rate, respiratory rate, oxygen saturation, body temperature, capillary refilling time, WBC count, and lactate level for logistic regression analysis. We determined the cut-off point for each continuous variable including age, body temperature, heart rate, respiratory rate, oxygen saturation, WBC count, and lactate value based on the ROC curve for each variable as follows: age ≤ 50 days, body temperature ≥ 38.5 °C, heart rate ≥ 178 bpm, oxygen saturation ≤ 96%, respiratory rate ≥ 50 breaths/min, WBC count > 14.0 × 10^3^, and lactate ≥ 3.7 mmol/L. In multiple logistic regression, male gender (Odds Ratio (OR) 2.45, 95% CI 1.41–4.28), body temperature ≥ 38.5 °C (OR 2.34, 95%CI 1.32–4.13), heart rate ≥ 178 bpm (OR 2.21, 95%CI, 1.28–3.83), and age ≤ 50 days (OR 2.13, 95%CI 1.21–3.77) were identified as significant predictors of serious infection (Table 4).

## 4. Discussion

This study aimed to evaluate the utility of the SIRS criteria evaluated at the ED admission in identifying serious infections in febrile young infants and demonstrated the limited impact of the SIRS criteria in identifying serious infections in infants younger than 90 days. The low to moderate values for the measures of diagnostic performance such as sensitivity (47.2%), specificity (68.0%), and LR+ (1.5) failed to provide clinicians with adequate information to choose appropriate acute care for febrile young infants. A study of adult sepsis showed that diagnostic performance measures such as sensitivity and specificity of the SIRS criteria were inferior to those of SOFA [25]. Furthermore, a recent neonatal study also demonstrated that SIRS criteria did not accurately identify late-onset sepsis in neonatal intensive care unit, with poorest accuracy in preterm infants [26]. In fact, the definition of SIRS in children was based on expert opinion because there was insufficient data on sepsis at that time [18].

Our study also identified clinical predictors of SI in febrile young infants that can be grouped under demographics and vital signs. In the demographic findings, we identified younger age (≤50 days), indicative of the immune system immaturity in young infants, as a predictive factor. Male gender was also an independent predictor, possibly due to the fact that uncircumcised male infants have a higher risk of urinary tract infection. Lack of circumcision can be a cause of SI because East Asian cultures normally do not circumcise newborns [27]. Among the vital sign variables included in SIRS criteria, heart rate and body temperature were found to be predictors, and the cut-off values calculated from the ROC curve in our model were close to those of the SIRS criteria. On the other hand, in line with the findings of a previous study of diagnosing pneumonia in children [28], the respiratory rate was less predictive for SI. Although the laboratory WBC count is being used as a low-risk criterion for predicting serious bacterial infections, our study indicated that the WBC count may not be useful for identifying SI. The number of patients with depressed WBC counts (<4000/uL) or elevated band (>10%) was small; thus, the impact of these variables in SIRS criteria might be limited. Instead, the percentages of neutrophils and lymphocytes were found to be valuable to screen the SI patients. The venous lactate value has been shown to be an effective predictor of pediatric severe sepsis in a large prospective study [29]; however; the study excluded infants younger than 60 days. Our study indicated that the venous lactate level may not be an effective predictor for SI at least in febrile young infants.

We therefore assumed that the low predictive value of the respiratory rate and WBC count in the SIRS criteria lowered the diagnostic performance of the SIRS criteria for identifying SI in young infants. Furthermore, it should be noted that patients’ demographic variables and vital signs are the most significant predictive factors because clinicians are able to obtain these data within the short period of time allowed for patient care in the ED.

### Limitations

This study has limitations. As this was a prospective study conducted at a single center in Japan, the etiology of the fever in young infants might differ from that in other regions, especially the Western countries. For example, newborns are rarely circumcised in Japan, with the result that the prevalence of urinary tract infections among young infants might be higher than in cultures which practice neonatal circumcision. However, pediatric societies in North America make allowances for the cultural background of the child and his family and do not routinely recommend circumcision anymore [30]. Next, we included the patients whose body temperature was 38.0 °C or higher; thus, it was possible to miss the sepsis patients with body temperature < 36.0 °C in SIRS criteria. In addition, since the false-negative rate of the rapid antigen test for influenza is high (approximately 30%) during the clinical course period, there is still a possibility of missing influenza cases in our cohort. Prospective, international studies are needed to verify the generalizability of our findings.

## 5. Conclusions

In conclusion, the SIRS criteria were ineffective in identifying serious infections in febrile infants younger than 90 days. However, such infections can be identified based on predictors such as age, gender, tachycardia, and high-grade fever.

## Figures and Tables

**Figure 1 children-08-01003-f001:**
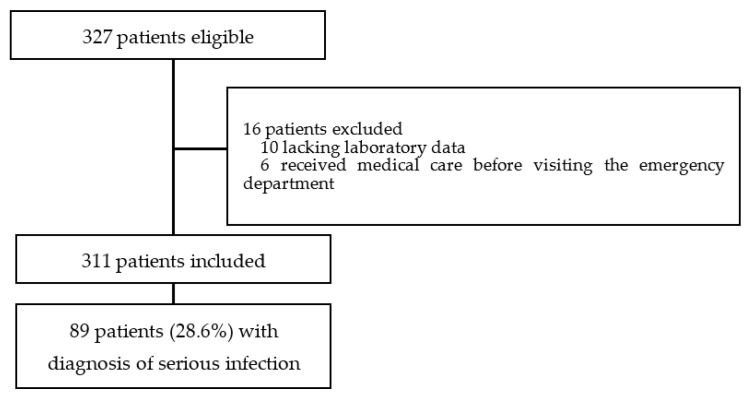
Derivation of study population.

**Table 1 children-08-01003-t001:** Clinical characteristics of patients (*n* = 311).

Variables	Mean (SD) or % (*n*)
Demographics	
Age, days	52.6 ± 23.7
Age	
0–27 days-old	19.3 (60)
28–60 days-old	37.0 (115)
61–89 days-old	43.7 (136)
Male	53.7 (167)
Gestational age, mean (SD), week	38.6 (1.5)
Birth weight, mean (SD), g, *n* = 310 (0.3% missing)	3008.8 (450.4)
Vital signs	
Core body temperature, mean (SD), °C	38.6 (0.5)
Heart rate, mean (SD), beats per minute	170.1 (19.5)
Respiratory rate, mean (SD), breaths per minute	38.1 (10.7)
Oxygen saturation, mean (SD), %	98.3 (2.2)
Physical examinations	
Peripheral poor perfusion, % (*n*)	40.8 (127)
Capillary refilling time ≥ 2 s, % (*n*)	22.8 (71)
Laboratory examinations	
WBC count, mean (SD), ×10^3^	10.5 (4.6)
Neutrophils percentage, mean, %	43.6
Lymphocytes percentage, mean, %	43.3
Monocytes percentage, mean, %	10.4
WBC > 12,000/μl, % (*n*)	92 (29.6)
WBC < 4000/μl, % (*n*)	7 (2.3)
Band > 10% with normal WBC, % (*n*)	5 (1.6)
Venous lactate (SD), mmol/L	2.7 (0.9)
Systemic Inflammatory Response Syndrome, % (*n*)	35.7 (114)
Clinical diagnosis	
Serious infections, % (*n*)	28.6 (89)
Serious bacterial infections, % (*n*) *	13.2 (41)
Serious viral infections, % (*n*) **	15.4 (48)
Disposition following the first visits	
Hospitalization in wards, % (*n*)	56.9(177)
Discharged from the emergency department, % (*n*)	40.8 (127)
Intensive care unit admission, % (*n*)	2.3 (7)
28-day mortality, % (*n*)	0 (0)

Note: SD, Standard Deviation; WBC, White Blood Cell. * Bacterial infections included 28 cases of urinary tract infection, 5 cases of bacteremia, 4 cases of bacterial pneumonia, 3 cases of soft tissue infection, and 1 case of meningitis patient. ** Serious viral infections included 14 cases of parechovirus infection (8 viremia and 6 meningitis), 14 cases of enterovirus infection (12 viremia and 2 meningitis), 14 respiratory cases of syncytial virus bronchiolitis, 4 cases of influenza virus pneumonia, and 2 cases of human metapneumovirus bronchiolitis.

**Table 2 children-08-01003-t002:** Performance of SIRS criteria in predicting serious infection.

	Sensitivity, % (95% CI)	Specificity, % (95% CI)	Positive Predictive Value, % (95% CI)	Negative Predictive Value, % (95% CI)	Positive Likelihood Ratio, (95% CI)	Negative Likelihood Ratio, (95% CI)
value (95%CI)	47.2 (36.6–58.0)	68.0 (61.4–74.0)	37.2 (28.4–46.8)	76.3 (69.6–81.9)	1.5 (1.1–2.0)	0.8 (0.6–0.9)

Note: CI, Confidence Interval.

**Table 3 children-08-01003-t003:** Comparison of patients with and without a serious infection.

Variables	Serious Infection (*n* = 89)	No Serious Infection (*n* = 222)	*p*
Age, days, Mean (SD)	49.2 (25.2)	54.1 (23.0)	0.101
Male, % (*n*)	68.5 (61)	47.7 (106)	0.001
Gestational age, mean (SD), week	38.6 (1.6)	38.6 (1.5)	0.796
Birth weight, mean (SD), g	3053.5 (421.6)	2990.5 (461.5)	0.253
Vital signs			
Core body temperature, mean (SD), °C	38.7 (0.5)	38.5 (0.4)	0.004
Heart rate, mean (SD), beats per minute	175.6 (20.9)	167.9 (18.4)	0.001
Respiratory rate, mean (SD), breaths per minute	38.8 (11.2)	37.8 (10.4)	0.429
Oxygen saturation, mean (SD), %	97.9 (3.2)	98.5 (1.6)	0.015
Physical findings			
Peripheral poor perfusion, % (*n*)	51.7 (46)	36.5 (81)	0.016
Capillary refilling time ≥ 2 s, % (*n*)	31.5 (28)	19.4 (43)	0.025
Laboratory examinations			
WBC count, mean (SD), ×10^3^	10.6 (4.9)	10.4 (4.5)	0.801
Neutrophils, mean, %	48.0	42.2	0.006
Lymphocytes, mean, %	39.3	44.5	0.008
Monocytes, mean, %	9.8	10.6	0.208
Venous lactate (SD), mmol/L	2.6 (0.9)	2.7 (0.9)	0.362

WBC, white blood cell; CRP, C-reactive protein.

**Table 4 children-08-01003-t004:** Adjusted risk of serious infection.

Characteristics	Odds Ratio	95% CI	*p*
Male	2.45	1.41–4.28	0.002
Core body temperature ≥ 38.5 °C	2.34	1.32–4.13	0.003
Heart rate ≥ 178 bpm	2.21	1.28–3.83	0.005
Age ≤ 50 days	2.13	1.21–3.77	0.009
Oxygen saturation ≤ 96%	1.89	0.92–3.87	0.083
WBC count ≥ 14.0 × 10^3^	1.49	0.76–2.92	0.249
Respiratory rate ≥ 50/min	1.44	0.71–2.95	0.316
Venous lactate ≥ 3.7 mmol/L	1.03	0.50–2.13	0.939

SD, standard deviation; WBC, white blood cell; CI, Confidence Interval.

## Data Availability

The data that support the findings of this study are available from the corresponding author upon reasonable request.

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
