# Peer review of "Limited Utility of SIRS Criteria for Identifying Serious Infections in Febrile Young Infants"

_children, 2021, doi:10.3390/children8111003_

Round 1
Reviewer 1 Report
The biggest conceptual flaw in the paper is reflected in the last line of the abstract: "A novel predictive approach for diagnosing serious infections in febrile young infants is needed." This flaw is reiterated throughout the paper. We already have a model for this that has been undergoing incremental improvement for 30 years. It is imperfect but undergoing continuous well-vetted scrutiny that is accepted by the AAP and ACEP.
The greatest strength of the paper is that it applies acceptable statistical analysis to affirm the doubts that most EM and PEM physicians have about using so-called SIRS criteria in infants.
A lesser conceptual flaw is reflected in line 53-54 regarding testing for viral infections: "is still difficult in the ED due to the length of time needed for viral detection using real-time polymerase chain reaction (PCR)." Length of time for testing is not the real issue. The real issue is that identifying the virus (e.g. entero or parecho) contributes little to identifying risk (i.e. most kids with entero or parecho do not go on to become septic). I agree with them (as does the current model) that viruses are important thus the change to "Serious Infection" rather than "Serious Bacterial Infection".
Likewise, lines 67-70 misrepresent most U.S. PEM physicians. We have never taken these references as anything more than just a suggested extrapolation from adult medicine to pediatrics. Most of us understand that there is no such thing as validated pediatric SIRS criteria, both in terms of how to define that criteria and how to use it.
Line 74 "there no consensus whether the sepsis criteria is useful to identify SI" is not quite right. There is consensus. Generally speaking, at least in the U.S., PEM and EM physicians regard elements of SIRS as clinical factors that can alert us and that we should consider, but not as elements that, in and of themselves, determine clinical decision-making.
Study Design: You need to acknowledge that your data acquisition pre-dates recent updates in the AAP/ACEP guidelines. Line 91: Please clarify - did you exclude patients who bounced-back and were admitted? That is probably the most important population that EM physicians worry about in regards to febrile infants.
Line 104: I understand why, for the sake of the study, that you took this approach. In reality, however, assessing sepsis risk is dynamic, and how well a patient responds to fluids and how sustained that response is, is part of that assessment. In particular, any patient who received fluids in the first 30 minutes of care must have looked sick, as it is uncommon for patients to receive fluids that quickly.
Lines 112-114: Most infants with soft tissue infection, pneumonia or UTI do better with algorithms that help identify low-risk, as these are the infant who traditionally have gotten too much work-up.
Lines 119-120: Many young infants with bronchiolitis receive these interventions, but that doesn't mean they have an illness that is comparable to sepsis.
Line 129: Flu rapid test had a false negative rate of 30% during that time period. You need to mention limitations such as that.
Line 233: The total WBC count does have limited usefulness, but what about band, lymphocyte and monocyte percentages? There is in fact evidence that total WBC is useful - see Bonsu and Harper.
Lines 242-245: There is nothing novel about considering age, gender, heart rate and temperature. Various combinations of these perform no better than SIRS criteria, the very approach that you are criticizing.
Overall, stick with your criticism of SIRS criteria, but avoid venturing into novel approaches - that is when you go beyond the scope of your paper. It is very easy to detect when a paper goes beyond its scope, and it turns readers off.
Author Response
Comment 1
“The biggest conceptual flaw in the paper is reflected in the last line of the abstract: "A novel predictive approach for diagnosing serious infections in febrile young infants is needed." This flaw is reiterated throughout the paper. We already have a model for this that has been undergoing incremental improvement for 30 years. It is imperfect but undergoing continuous well-vetted scrutiny that is accepted by the AAP and ACEP.”
Response
Thank you for your important pointing out. As you suggested, the description regarding the new models identifying serious infections was misleading. We removed that description regarding the new predictive approaches in the abstract, discussion, and conclusion.
Comment 2
“The greatest strength of the paper is that it applies acceptable statistical analysis to affirm the doubts that most EM and PEM physicians have about using so-called SIRS criteria in infants.”
Response
Thank you for your positive comments. We focused on arguing the performance of SIRS criteria.
Comment 3
“A lesser conceptual flaw is reflected in line 53-54 regarding testing for viral infections: "is still difficult in the ED due to the length of time needed for viral detection using real-time polymerase chain reaction (PCR)." Length of time for testing is not the real issue. The real issue is that identifying the virus (e.g., entero or parecho) contributes little to identifying risk (i.e. most kids with entero or parecho do not go on to become septic). I agree with them (as does the current model) that viruses are important, thus the change to "Serious Infection" rather than "Serious Bacterial Infection"”.
Response
Thank you for your critical comments. As you pointed out, the real issue is that most of the infants with entero or parecho virus are not septic. Accordingly, we removed the statement about PCR and stated the issue in the real world for seeing the patients with those infections (Line 55-55).
Comment 4
“Likewise, lines 67-70 misrepresent most U.S. PEM physicians. We have never taken these references as anything more than just a suggested extrapolation from adult medicine to pediatrics. Most of us understand that there is no such thing as validated pediatric SIRS criteria, both in terms of how to define that criteria and how to use it.”
Response
Thank you for letting us know that the current practice of PEM physicians in the U.S. We have corrected this description in response to your suggestion (Line 67-71).
Comment 5
“Line 74 "there no consensus whether the sepsis criteria is useful to identify SI" is not quite right. There is consensus. Generally speaking, at least in the U.S., PEM and EM physicians regard elements of SIRS as clinical factors that can alert us and that we should consider, but not as elements that, in and of themselves, determine clinical decision-making.”
Response
Thank you for your suggestion. In response to your comment, we removed the descriptions on the consensus of SIRS criteria in practice for febrile young infants.
Comment 6
“Study Design: You need to acknowledge that your data acquisition pre-dates recent updates in the AAP/ACEP guidelines. “
Response
Accordingly, we added that this study was conducted before the AAP/ACEP guideline update (Line 89-91).
Comment 7
“Line 91: Please clarify - did you exclude patients who bounced-back and were admitted? That is probably the most important population that EM physicians worry about in regards to febrile infants.”
Response
Our apology for the confusing description. No, we did not exclude the patients who bounced-back and were admitted. We corrected the description to clarify this point (Line 92-93).
Comment 8
“Line 104: I understand why, for the sake of the study, that you took this approach. In reality, however, assessing sepsis risk is dynamic, and how well a patient responds to fluids and how sustained that response is, is part of that assessment. In particular, any patient who received fluids in the first 30 minutes of care must have looked sick, as it is uncommon for patients to receive fluids that quickly.”
Response
We agree with you. Seventeen patients (5.4%) received fluids bolus in the first 30 minutes of care, and all of them showed any signs of sepsis (e.g., tachycardia, impaired perfusion, and prolonged CRT) in the initial assessment before the bolus intervention. Thus, we consider that our approach did not influence the study results. Thank you for your comment.
Comment 9
Lines 112-114: Most infants with soft tissue infection, pneumonia or UTI do better with algorithms that help identify low-risk, as these are the infant who traditionally have gotten too much work-up.
Lines 119-120: Many young infants with bronchiolitis receive these interventions, but that doesn't mean they have an illness that is comparable to sepsis.
Response
We agree with the reviewer's comment. As you indicated, most infants with these infections are stable. However, we consider that these infections need to be included as the primary outcomes (i.e., serious infections) in this study as the infants with these bacterial infections require antibiotic treatment.
Regarding bronchiolitis, we previously reported a paper on serious viral infection of young infants in the Journal of Paediatrics and Child Health (Hayakawa et al., J Paediatr child health 2020, 56, 586-589) and used this definition for "severe bronchiolitis." We consider that “severe bronchiolitis” is a critical illness that should be screened at ED because it can be life-threatening if left untreated. Thank you for your comments.
Comment 10
Line 129: Flu rapid test had a false negative rate of 30% during that time period. You need to mention limitations such as that.
Response
We commented on this issue in the limitations (Line 272-275).
Comment 11
Line 233: The total WBC count does have limited usefulness, but what about band, lymphocyte and monocyte percentages? There is in fact evidence that total WBC is useful - see Bonsu and Harper.
Response
Thank you very much for your insightful suggestions. We statistically analyzed the differences in the lymphocytes and monocytes and found the statistical difference in the percentages between the SI and non-SI groups. We added the description of this new finding in the result and discussion sections (Line 190-193, 242-243).
Comment 12
“Lines 242-245: There is nothing novel about considering age, gender, heart rate and temperature. Various combinations of these perform no better than SIRS criteria, the very approach that you are criticizing.”
Response
In response to the reviewer’s comment, we deleted this description.
Comment 13
“Overall, stick with your criticism of SIRS criteria, but avoid venturing into novel approaches - that is when you go beyond the scope of your paper. It is very easy to detect when a paper goes beyond its scope, and it turns readers off.”
Response
Thank you for your important suggestion. We removed all the descriptions regarding the novel approaches in this manuscript and focused on criticizing the SIRS criteria.
Reviewer 2 Report
The aim of the study was to evaluate the diagnostic performance of the SIRS criteria for identifying serious infections (in Section 2.4. defined as septicemia, meningitis, pneumonia etc.) in febrile infants < 90 days. The same definition of the aim of the study is repeated in Discussion. However, the title of the manuscript is referring to septic young infants. Can the authors elaborate why they chose this title?
Authors state: „For this study, we adopted the definition of SIRS provided by the International Pediatric Sepsis Conference Guideline [18].“ One of the criteria is: Core temperature of >38.5°C or <36°C. Since only the infants with temperature >38 were included in the study, is there a possibility that some infants with SI were missed?
The authors enrolled 311 patients out of which 114 met the SIRS criteria. The vitals and the SIRS criteria were evaluated at the admission to the emergency department and, in my opinion that should be clearly stated throughout the manuscript.
Total of 89 patients had a serious infection. When was SI diagnosed, at admission in all of them or during the course of disease?
Please check text in Fig. 1. In my version of the manuscript, it seems like some words are missing.
Also, please check and verify number of patients in whom gestational age and birth weight were collected (Table 1).
In Table 1., the data for Age is presented twice. One may be deleted.
Line 179-180: …SI patients presented a significantly higher body temperature (38.7 [0.5] vs 38.7 [0.5], p=0.004)… Please check and verify if this is accurate.
According to the reference 18, one of the SIRS criteria is: Leukocyte count elevated or depressed for age or >10% immature neutrophils. The authors presented mean WBC count in Table 3. Were there any patients with depressed count? What about immature neutrophils: were they evaluated as well? I suggest to address this issue in Discussion.
Author Response
Comment1
“The aim of the study was to evaluate the diagnostic performance of the SIRS criteria for identifying serious infections (in Section 2.4. defined as septicemia, meningitis, pneumonia etc.) in febrile infants < 90 days. The same definition of the aim of the study is repeated in discussion. However, the title of the manuscript is referring to septic young infants. Can the authors elaborate why they chose this title?”
Response
Thank you for your insightful advice. We change the title to "Limited Utility of SIRS Criteria for Identifying Serious Infections in Febrile Young Infants."
Comment 2
“Authors state: "For this study, we adopted the definition of SIRS provided by the International Pediatric Sepsis Conference Guideline [18]." One of the criteria is: Core temperature of >38.5°C or <36°C. Since only the infants with temperature >38 were included in the study, is there a possibility that some infants with SI were missed?”
Response
Thank you for your critical comments. As you indicated, the infants with serious infections might have been missed if their body temperature was lower than 38.0 °C. It was anticipatory, but we had to design like this for the sake of the study feasibility. We added the description regarding this issue in the limitation section of the discussion (Line 270-272).
Comment3
“The authors enrolled 311 patients out of which 114 met the SIRS criteria. The vitals and the SIRS criteria were evaluated at the admission to the emergency department and, in my opinion that should be clearly stated throughout the manuscript.”
Response
We clarified that the SIRS criteria were evaluated at the ED admission (Line 77, 100-101, 156, 212-213).
Comment 4
Total of 89 patients had a serious infection. When was SI diagnosed, at admission in all of them or during the course of disease?
Response
Thank you for your question. The patients were diagnosed during the disease as the diagnosis can change in the clinical course in some patients. We clarified this point in the methods (Line 101-103).
Comment 5
“Please check text in Fig. 1. In my version of the manuscript, it seems like some words are missing.
Also, please check and verify a number of patients in whom gestational age and birth weight were collected (Table 1).
In Table 1., the data for Age is presented twice. One may be deleted.
Line 179-180: …SI patients presented a significantly higher body temperature (38.7 [0.5] vs 38.7 [0.5], p=0.004)… Please check and verify if this is accurate.”
Response
Thank you for your careful checking. We corrected the point you kindly suggested (Figure1, Table1, Line 185).
Comment 6
According to the reference 18, one of the SIRS criteria is: Leukocyte count elevated or depressed for age or >10% immature neutrophils. The authors presented mean WBC count in Table 3. Were there any patients with depressed count? What about immature neutrophils: were they evaluated as well? I suggest to address this issue in discussion.
Response
Thank you for your important queries. The number of patients with WBC<4,000/μl, and band >10% with normal WBC count was 7 and 5, respectively. These variables were taken into consideration in the evaluation for meeting SIRS criteria. We added these in the table and the descriptions in the discussion as you suggested (Table 1, Line 240-243).
Round 2
Reviewer 1 Report
This is better. I'd recommend mild revisions in regards to making the language even more concise. This is not essential but is generally appreciated by readers.
In Line 89, add the word "updated". You are generally aligned with the guidelines, it's just the latest updates that came in after your study.
There may very well be challenges to your paper. Be prepared for letters that present different ways of interpreting your data or statistics.
Author Response
Thank you very much for your supportive comments.
Comment 1
" I'd recommend mild revisions in regards to making the language even more concise. This is not essential but is generally appreciated by readers."
Response:
We simplified several redundant points to improve the readability of the manuscript. Please see the revised paper.
Comment 2
"In Line 89, add the word "updated". You are generally aligned with the guidelines, it's just the latest updates that came in after your study."
Response:
We added "updated" at the suggested point. Thank you for your careful reading.
Comment 3
"There may very well be challenges to your paper. Be prepared for letters that present different ways of interpreting your data or statistics."
Response:
Thank you for your thoughtful advice. We would like to discuss with the readers to learn further in this important topic.